# The Influencing Factors of Pro-Environmental Behaviors of Farmer Households Participating in Understory Economy: Evidence from China

Yaru Chen [1,2], Xiao Han [2,*] , Siyao Lv [2], Boyao Song [2], Xinye Zhang [1] and Hongxun Li [2]

1   Development Research Center, National Forestry and Grassland Administration, Beijing 100714, China
2   School of Economics and Management, Beijing Forestry University, Beijing 100083, China
*   Correspondence: hangood@bjfu.edu.cn; Tel.: +86-10-17801008030

**Abstract:** To promote the sustainable development of agriculture and forestry economy, it is of great significance to guide farmers to consciously pursue pro-environmental behaviors in the development of the understory economy. Based on field survey data from Yong'an city of Fujian Province and Luoshan County of Henan Province, factor analysis and Oprobit models were mainly used to analyze the influencing factors and influencing the degree of the pro-environmental behaviors of farmer households participating in the understory economy. Quantitative data showed that individual characteristics of farmers (i.e., gender, education, career, feasibility evaluation, and the proportion of farming labor to household labor) and forest land management status (i.e., forest land transfer, the working time in understory economy, and proportion of understory economic income) have an obvious effect on the adoption of pro-environmental behaviors. Findings also revealed three key variables (i.e., the farmers' environmental perception, social constraints, and government incentives) that are associated with the willingness to adopt pro-environmental behaviors. The contribution ratios of the influencing factors were environmental perception, social constraints, and government incentives. In addition to economic benefits, perceptual factors and informal institutions also play an important role in driving farmers to adopt pro-environmental behaviors. Based on the findings, it is necessary to strengthen the publicity and the education of farmers, increase environmental responsibility, accelerate the establishment and improvement of ecological reward-and-punishment mechanisms, and enhance the training of green production techniques.

**Keywords:** understory economy; sustainable development; pro-environmental behavior; influencing factors; influencing degree

## 1. Introduction

Understory economy relies on forest resources and forest ecological environment and enables farmers to make full use of their forests while managing them in a scientific manner [1]. It is known as a sustainable forest management mode that can provide income for farmers without having to cut trees. For example, farmer households can breed poultry, plant Chinese medicinal species, and cultivate mushrooms in their forests [2]. Such production management mode is a three-dimensional compound mode, which, as an ecological agriculture mode, can realize the sharing of farming, forestry, husbandry, and fishing resources, the complementary advantages, the circulation and intergeneration, and the coordinated development of various agricultural production systems [3]. Furthermore, understory economy is crucial in the current global context of climate change. It will increase the resilience of both rural and urban municipalities, as well as promote local economies. However, at present, there is no internationally accepted concept of an understory economy. In a broad sense, the concept of understory economy is similar to that of agroforestry [4]. Nevertheless, in contrast to agroforestry, the understory economy is not an utterly industrial operation mode but emphasizes the principle of "ecological

priority", which is a unique idealized development concept first proposed by the Chinese government [5].

Harmony between humans and their ecological environment is a global issue, and the relationship between economic development and environmental protection has been discussed for decades [6]. It is generally believed that economic development entails environmental damage, but a great natural environment and opportunities to get close to nature will promote a sense of happiness [7,8]. Chairman Xi of China proposed the development idea of "Lucid waters and lush mountains are gold and silver mountains", which made the Chinese people pay more attention to the protection of the ecological environment and the sustainable use of resources in production and management activities. The purposes of developing an understory economy are to promote economic growth; improve the livelihood of forest farmers; consolidate the achievements of poverty alleviation; make up for the inherent shortcomings of forestry, such as long operation cycles; and obtain economic and social benefits on the premise of ensuring the ecological benefits of forests without destroying their ecological resources [1].

Currently, in China, the development of an understory economy not only brings economic benefits but also produces certain environmental problems [9]. For example, although chemical fertilizers and pesticides can promote crop growth to some extent [10], their excessive use has caused agricultural non-point source pollution [11,12]. China is a major user of chemical fertilizers and pesticides [13]. In 2021, the amount of chemical fertilizer applied per hectare of cropland in China reached 506.11 kg, which was 2.05 times that of the UK and 3.69 times that of the USA. The number of pesticides used in China's crops was 10.3 kg/hectare, which was considerably above the global average value, resulting in a decline in the quality of cultivated land. In agricultural production activities, if plastic agricultural films and other wastes are not treated properly, it causes great pressure on the ecological environment [14]. To pursue short-term profits, farmer households often ignore the regional ecological carrying capacity and blindly develop and reclaim agricultural resources. The high regional planting and cultivation density leads to the consumption of large amounts of groundwater, the decline of woodland plant diversity, and the decrease in ecological functions, threatening ecosystem stability [15].

Farmer households play an important role in understory economic activities [16]. When farmer households participate in the production and management of an understory economy, they mainly pursue short-term economic benefits [17]. In addition, due to the lack of a collective sense of efficacy, farmers are worried that others are not able to effectively regulate their production behaviors, whereas their own actions may increase the environmental costs [18,19]. As a result, they also refuse to adopt pro-environmental production behaviors. The understory economy industry is an important bridge between "lucid waters and lush mountains" and "gold and silver mountains", and the primary prerequisite for the development of the understory economy envisioned by the government is to ensure that forests are not destroyed. Therefore, the goals of farmer households and the government for understory economic activities are not exactly the same. To promote the sustainable development of the understory economy and ensure ecological benefits, it is necessary for farmer households to adopt environmentally friendly behaviors pointing to common welfare in the production process [20]. On the one hand, environmentally friendly behaviors can reduce the cost of environmental governance, and on the other hand, they can facilitate the healthy development of the understory economy industry [21]. Therefore, it is of great importance to guide farmer households to take environmental protection behaviors when participating in an understory economy.

The study aims to fill the gap by exploring the specific factors that affect farmer households' pro-environmental behaviors when participating in an understory economy. In this paper, based on the current situation in China, starting from the microscopic perspective, such as the economics theories and then farmer household behavior theory, we established an evaluation indicator system of the pro-environmental behaviors in line with the particularity of the development mode of understory economy and then conducted a questionnaire

survey. In order to maintain the original purpose of developing the understory economy, which is to make ecological benefits a priority rather than economic benefits, this paper attempts to explore sustainable ways to minimize the negative influence and maximize the positive impact by guiding farmers' production behavior. In this regard, we elaborate on the following questions.

- What kind of behavior is pro-environmental in the process of understory economy development, and what is the current level of pro-environmentality of farmers?
- What factors can effectively influence the level of pro-environmental behavior of farmers?
- What is the probability that these influences can play a role in increasing the level of pro-environmental behavior of farmer households?

According to the results, we propose some suggestions aiming to reduce the negative impacts of understory economic development on the ecological environment and to guide farmer households to adopt pro-environmental behaviors in their production activities. Our work has crucial practical and theoretical significance for promoting China and other developing countries to achieve a balance between economic development and environmental protection.

## 2. Research Framework

The concept of pro-environmental behaviors was put forward in the 1980s in the field of psychology [22]. However, with the continuous improvement of social awareness in terms of ecological environment protection, pro-environmental behaviors have begun to develop and be applied to various fields. The core idea of such behaviors is to adopt environmentally beneficial behaviors in the behavioral process of various fields. Pro-environmental behaviors, first proposed by Hines et al., were defined as conscious actions to protect the ecological environment based on an individual's sense of social responsibility and values. It is a typical behavior beneficial to society, which can play a role in safeguarding the basic interests of individuals and organizations and is conducive to environmental protection and social development [23,24]. In terms of research on the influences of pro-environmental behaviors, Brick et al., focusing on the reduction of greenhouse gas emissions, studied the factors influencing pro-environmental behaviors and highlighted openness, agreeableness, urbanization, and environmental awareness [25]. Kim et al. tested the influences of altruistic values and attitudes on pro-environmental behaviors by using the value–attitude–behavior model [26]. According to the study of Arezu Shafiei et al., people would decide whether to adopt pro-environmental behaviors by comparing the costs and benefits of current behaviors and pro-environmental behaviors; that is, the costs and benefits of pro-environmental behaviors will affect the decision for pro-environmental behaviors [27]. Based on the theory of planned behavior, Valizadeh et al. explored how subjective norms and self-concept can effectively enhance farmers' intentions toward environmental protection [28]. Through a study of farmers' green technology adoption, Xie H. and Huang Y. found that the positive impact of land transfer on farmers' adoption of pro-environmental behaviors was not affected by the time [29].

As an important subject involved in the understory economic industry, the farmer household is the direct decision maker of production and management activities, and its behavior affects the development status of the understory economic industry. According to the theory of farmer household behaviors, farmer households not only have rational decision behaviors as economic means but also own their special natures as the basic social group. Due to influencing factors such as limited cognition, environmental impact, and information asymmetry, farmer households can only realize "bounded rationality" when making decisions in an understory economy. Therefore, the behavioral decision of farmer households is the maximum value of the utility function under the constraints of the rationality degree, the utility of their own resources, and the time consumed in understory production [30]. It is affected by their own resource endowment, the cognition of farmer households on environmental protection and the economic industry under the forest, the

conditions of the farmers' access to information, and environmental factors such as society and government [31]. Furthermore, by combination with the behavioral attitude, subjective norms, and perceived behavioral control ability in the theory of planned behaviors, this study divides the factors that affect the decision making of farmer households into two categories: internal and external factors [32]. The internal factors include the individual characteristics of farmer households, forest land management, and farmer households' perception of environmental protection. The external factors include social constraints such as rural culture, regional characteristics, and government incentives. Figure 1 shows the established theoretical framework for influencing farmer households' participation in pro-environmental behaviors.

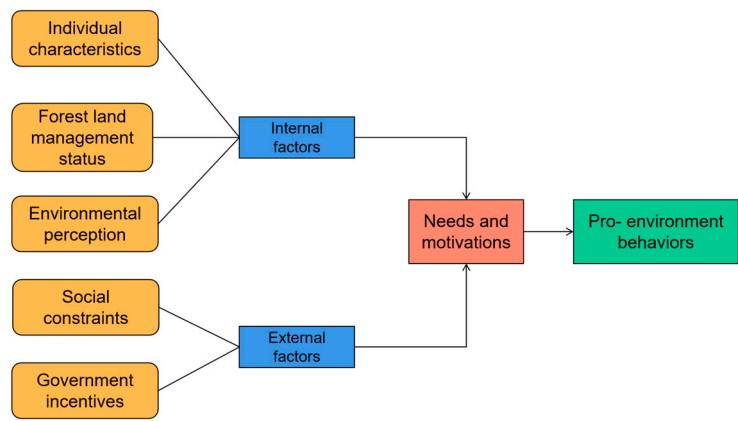

**Figure 1.** Research framework.

## 3. Methods

### 3.1. Variable Selection

#### 3.1.1. Dependent Variables

Based on three links of pre-production, mid-production, and post-production in understory economic production activities, this study divides the pro-environmental behaviors of farmer households in the production process into a total of ten pro-environmental behaviors (as listed in Table 1), including three typical modes of understory planting, understory breeding, and the combination of these two modes.

**Table 1.** Measurement of farmers' pro-environmental behaviors in understory economy.

| Understory Economy Mode | Indicators of Dependent Variables |
|---|---|
| The combination of understory planting and breeding | Whether the forest clearing activities are taken (Y1)<br>Whether the planting and breeding varieties are pro-environmental (Y2)<br>Whether planting and breeding density are strictly controlled (Y3)<br>Whether enclosure and fence are used on the forest land (Y4) |
| Understory breeding | Whether cleaning the shed of livestock breeding is frequently carried out (Y5)<br>Whether the treatment of dead poultry and livestock is pro-environmental (Y6)<br>Whether disposing of livestock manure and other wastes is sustainable |
| Understory planting | Whether water-saving methods are used for irrigation (Y8)<br>Whether the usage of chemical fertilizers and pesticides is pro-environmental (Y9)<br>Whether the treatment of plastic sheeting is pro-environmental (Y10) |

#### 3.1.2. Independent Variables

According to theoretical analysis, the pro-environmental behaviors of farmer households when participating in understory economic production and management are mainly affected by the individual characteristics of farmer households, the forest land management status, the farmer households' environmental perception, social constraints, and government incentives. Among these factors, the core independent variables include farmer

households' environmental perception, social constraints, and government incentives, and the control variables include individual characteristics and forest land management of farmer households.

- Farmer households' environmental perception

Environmental perception is an important factor influencing farmer households' pro-environmental behaviors [33]. It refers to believing in, understanding, and evaluating environmental-protection-related issues during forestry production in understory economic activities. According to cognitive behavior theory, individuals can develop environmental awareness only after mastering certain environmental knowledge [34]. The smaller the deviation of the knowledge of pro-environmental behaviors mastered by farmer households and the more scientific the acquired knowledge of pro-environmental behaviors, the stronger the impetus that can support their adoption of pro-environmental behaviors [35]. Therefore, in the context of understory economic activities, the higher the farmer's awareness of environmental protection, the more pronounced the mobilization of the inner subjective initiative for pro-environmental behaviors, and, thus, the higher the degree of participation in pro-environmental behaviors [36]. In addition, the higher the evaluation of economic and ecological benefits and the lower the evaluation of costs, the stronger the willingness of farmers to adopt pro-environmental behaviors [37].

- Social constraints

Social constraints are some of the external situational factors for farmer households to adopt pro-environmental behaviors [38]. China's rural society is still a relational society; that is, the relationships among farmer households depend on human favor, individual prestige, and a relationship network constructed in the form of a "differential pattern" [39]. Therefore, when discussing the factors influencing farmers' pro-environmental behaviors, it is necessary to take the social constraints faced by farmers into account, which mainly include reputation appeal, place attachment, parents' experience, group pressure, key man effect, and social trust [40]. First, according to the reputational utility theory, good interpersonal relationships and being respected by others are the internal pursuits of human beings, and reputation can bring considerable satisfaction to behavioral subjects [41]. The more attention the farmer households pay to reputation, the stronger their motivation to adopt pro-environmental behaviors [42]. Second, against the unique cultural environment background of China, farmers have a stronger sense of belonging to the land, which can generate psychological ownership for the region; that is, an individual "sense of ownership" can enhance the cherished attitude to the land, making "place attachment" become an important factor affecting farmers' pro-environmental behaviors [43]. The stronger the place attachment, the more likely the farmers are to adopt pro-environmental behaviors. Moreover, the "close" social relations in rural areas have established a relatively stable social network. Observation learning is one of the main ways of social learning for farmer households, and the action choice of others can greatly affect individual judgment [44]. When faced with choices, farmer households are often pushed by individuals or relevant organizations and take others' behavior choices as the reference standard. For example, their understory production and management behaviors may refer to their parents, surrounding relatives and neighbors (groups), and rural elites (key people). When other farmer households in the same group adopt pro-environmental behaviors, those who have the intention but do not take action will feel guilty and move closer to the group behaviors [45].

- Government incentives

Government incentives can also significantly influence the external scenario of farmer households adopting pro-environmental behaviors. Generally, the environment has the attribute of a public good [46]. Since the external intervention is a necessary way to encourage farmer households to adopt pro-environmental behaviors, the government can influence farmers' pro-environmental behaviors through subsidies and guidance [47]. On the one hand, subsidies directly encourage pro-environmental behaviors by increasing income, and

ecological compensation may enhance the motivation for pro-environmental behaviors. On the other hand, through technical training, the government can help farmers overcome the obstacles encountered in the process of adopting pro-environmental behaviors, increase their environmental knowledge, and reduce the cost of pro-environmental behaviors, thus promoting the adoption of pro-environmental behaviors in understory production and management [48].

- Individual characteristics and forest land management of farmer households

This study investigated two control variables: individual characteristics of farmer households and their forest land management status. The individual characteristics include gender, age, education level, occupation, the proportion of the agricultural labor force, and whether there are village cadres in the family. The forest land management status involves whether the forest land is transferred, the average distance from the home to the forest land, the working time in the understory economy, and the proportion of the annual economic income from the understory economy in the total household income.

Based on the above analysis, and by combination with relevant literature and the outcomes of the questionnaires, we selected a total of 24 variables from 5 aspects (see Table 2 for details) to further discuss the factors influencing farmers' pro-environmental behaviors when participating in an understory economy.

**Table 2.** Indicator description and expectation.

| Grade 1 | Grade 2 | Indicator Description | Expectation |
|---|---|---|---|
| Environmental perception | Environmental awareness ($P_1$) | 1–5 indicates different degrees of recognition. The higher the value, the higher the degree of recognition. | + |
| | Perception of functions of pro-environmental behavior ($P_2$) | | + |
| | Perception of benefits of pro-environment behaviors ($P_3$) | | + |
| | Perception of responsibility for pro-environmental behaviors ($P_4$) | | + |
| | Perception of cost of pro-environmental behaviors ($P_5$) | | − |
| | Perception bias of pro-environmental behaviors ($P_6$) | | +/− |
| Social constraints | Reputation appeal ($S_1$) | 1–5 indicates different degrees of recognition. The higher the value, the higher the degree of recognition. | + |
| | Place attachment ($S_2$) | | + |
| | Parents' experience ($S_3$) | | +/− |
| | Group pressure ($S_4$) | | +/− |
| | Key man effect ($S_5$) | | +/− |
| | Social trust ($S_6$) | | + |
| Government incentives | Ecological compensation ($G_1$) | 0 = No, 1 = Yes. | + |
| | Government guidance ($G_2$) | 1 = Never, 2 = Once a year or more, 3 = Once half a year, 4 = Once a quarter,5 = Once less than a quarter. | + |
| Individual characteristics | Age | 1 = Less than 30, 2 = 31–45, 3 = 46–55, 4 = 56–70, 5 = Above 70. | +/− |
| | Gender | 0 = Female, 1 = Male. | +/− |
| | Education | 1 = Illiteracy, 2 = Primary school diploma, 3 = Junior high school diploma, 4 = High school or technical secondary school diploma, 5 = University degree, college degree or above. | + |
| | Career | 1 = Farmer, 2 = Others. | +/− |
| | The proportion of farming labor to household labor | The number of farming labor/The number of household labor. | + |
| | Village cadres at home | 0 = No, 1 = Yes. | + |

**Table 2.** *Cont.*

| Grade 1 | Grade 2 | Indicator Description | Expectation |
|---|---|---|---|
| | Whether forest land is transferred | 0 = No, 1 = Yes. | +/− |
| Forest land management status | The average distance of forest land from home | 1 = 0–1 km, 2 = 1 km–5 km, 3 = Above 5 km. | − |
| | The working time in understory economy | Years of under-forest economy. | + |
| | Proportion of understory economic income | Understory economic income/Total family income. | +/− |

*3.2. Main Model*

The dependent variable (the pro-environmental behavior) was a typical discrete multi-classification ordinal variable; thus, the Oprobit model was selected for the quantitative analysis of influencing factors, which can be expressed as follows:

$$y* = \sum_{i=0}^{j} \beta_i X_i + \varepsilon, \varepsilon | X \sim N(0,1) \tag{1}$$

$$\begin{cases} y* \leq \alpha_1, \ y = 0 \\ \alpha_1 < y* \leq \alpha_2, \ y = 1 \\ \cdots \cdots \\ \alpha_j < y*, y = j \end{cases} \tag{2}$$

The probability of the *i*-th value of *y* locating in the scope of *j* can be expressed as follows:

$$\begin{aligned} P(y_i = 1) &= P(y^* < \alpha_1) = \varnothing (\alpha_1 - \beta_i X_i) \\ P(y_i = 2) &= P(\alpha_1 < y^* < \alpha_2) = \varnothing (\alpha_2 - \beta_i X_i) - \varnothing (\alpha_1 - \beta_i X_i) \\ &\cdots \cdots \\ P(y_j = j) &= P(\alpha_{j-1} < y^* < \alpha_j) = \varnothing (\alpha_j - \beta_i X_i) - \varnothing (\alpha_{j-1} - \beta_i X_i) \end{aligned} \tag{3}$$

where $y*$ is the latent variable; $y$, which takes the value of 1, 2, 3, or 4, represents the comprehensive level of the farmer households adopting pro-environmental behaviors, and a larger value of $y$ indicates a better implementation of the pro-environmental behaviors; $X$ represents the factor influencing farmers' participation in pro-environmental behaviors in understory economy; $\beta$ is the independent variable coefficient; $\varepsilon$ is the error term; $\alpha_j$ is the parameter to be estimated; $\varnothing(.)$ is the standard normal distribution function.

In addition, the regression results of the ordinal logit model were also compared with the estimation results of the ordinal probit model to verify the robustness of the estimation results.

**4. Data and Descriptive Analysis**

*4.1. Data Collection*

Yong'an city (county-level city) of Fujian Province and Luoshan County of Henan Province were selected as the research areas. The two counties (cities) have rich forest resources and a good development status of the forestry industry. Yong'an City is located in the west of the central part of Fujian Province, covering a total area of 2931 square kilometers. As a mountainous area, it has the topographic characteristics of "nine hills, one-half water, and one-half field". The forest coverage rate is 82.85%, and the forest stock is 27.16 million cubic meters. Since 2014, Yong'an City has been recognized as the "National Understory Economic Demonstration Site". The area of understory planting, mainly for planting traditional Chinese medicinal crops, covers 30.3 square kilometers, with an annual output value of about CNY 360 million. The area of understory breeding, mainly for raising chickens, geese, and frogs, covers 28.3 square kilometers, and the output value is about CNY 410 million. Luoshan County is located in the south of Henan, at the north foot of Dabie Mountain and the south bank of the Huaihe River. It has a large topographic span,

with a total area of 2077 square kilometers and a forest coverage rate of 39.92%. By the end of 2019, the development scale of the understory economy of Luoshan County had reached CNY 1 billion, and the area of the understory economy industry had reached 263 square kilometers. Nowadays, there are four major mature understory economy types in Luoshan County, namely understory collection, understory tourism, understory culture, and understory planting.

In November 2020 and May 2021, our research group visited Yong'an and Luoshan County, respectively, to conduct questionnaire surveys and face-to-face interviews. A total of 257 questionnaires were distributed, among which 246 were valid, with an effective rate of 95.7%. The specific geographical location of the study area is shown in Figure 2.

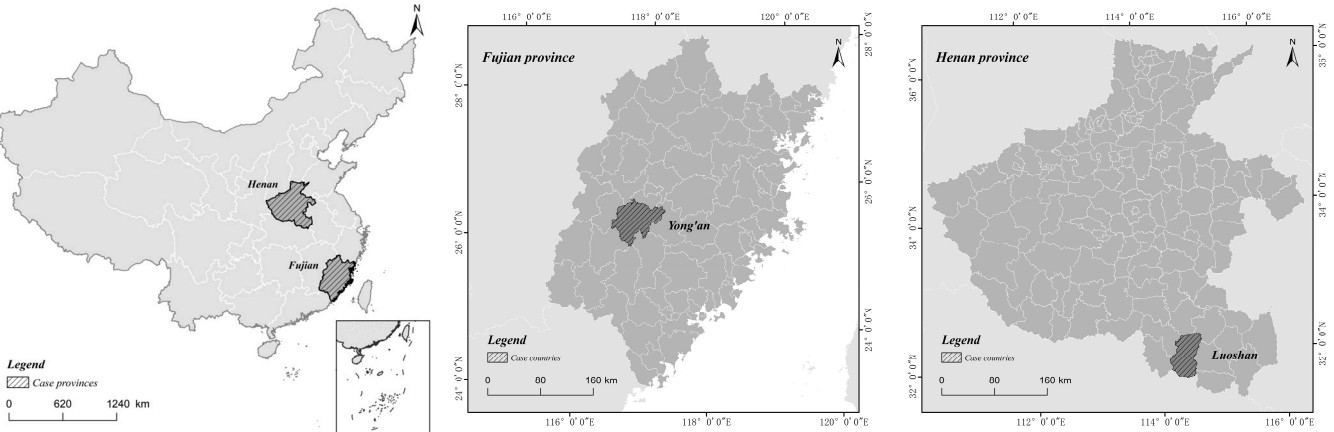

**Figure 2.** Location of provinces and countries.

*4.2. Descriptive Analysis*

- Farmer households' pro-environmental behaviors

The obtained data passed the Kaiser–Meyer–Olkin measure of sampling adequacy (KMO) test and Bartlett's sphericity test criteria. The KMO values were 0.536 and 0.466, respectively, and the *p* value of Bartlett's sphericity test was 0.000. Subsequently, factor analysis and principal components analysis were applied to measure the level of pro-environmental behaviors of farmer households participating in the understory economy and to assign different levels with appropriate values. The score ranges of the principal component were <−0.5, −0.5–0, 0–0.5, and >0.5, which, respectively, represent low, relatively low, relatively high, and high levels of pro-environmental behaviors. The specific distribution of the implementation levels is shown in Table 3.

**Table 3.** Level distribution of implementation of pro-environmental behaviors.

| Levels of Pro-Environmental Behaviors | Score Range of Principal Components | Frequency | Percentage |
|---|---|---|---|
| Poor | −1.43~−0.5 | 40 | 16.26% |
| Fair | −0.5~0 | 101 | 41.06% |
| Good | 0~0.5 | 73 | 29.67% |
| Excellent | 0.5~1.32 | 32 | 13.01% |

- Basic characteristics and forest land management status of farmer households

We surveyed 81 understory breeding households and 165 understory planting households. Through interviews with farmers, we found out that the main reasons for the selection of different understory economic modes are the cost. The cost of understory breeding is much higher than that of understory planting, even though the return is relatively high. Therefore, under the influence of insufficient initial funds, most farmers

choose understory planting. Male heads of households accounted for 90.7% of the surveyed households, and the median ages were between 56 and 70 years old, which is in line with the real situation in rural China. The main participants in the understory economy showed remarkable aging characteristics. As many as 41.1% of the rural households had a low education level (only primary school education). Further, 51.2% of the surveyed households were in the part-time working state; young and middle-aged members in most household families migrated out for work, whereas the old people were engaged in understory economic production activities (see Table 4 for details).

**Table 4.** Sample Distribution Characteristics.

| Variable | Values | Frequency | Percentage |
|---|---|---|---|
| Gender | Male | 233 | 90.7% |
| | Female | 23 | 9.3% |
| Age | 1 = Less than 30 | 4 | 1.6% |
| | 2 = 31–45 | 18 | 7.4% |
| | 3 = 46–55 | 95 | 38.6% |
| | 4 = 56–70 | 107 | 43.5% |
| | 5 = Above 70 | 22 | 8.9% |
| Education | 1 = Illiteracy | 14 | 5.7% |
| | 2 = Primary school diploma | 101 | 41.1% |
| | 3 = Junior high school diploma | 62 | 25.2% |
| | 4 = High school or technical secondary school diploma | 66 | 26.8% |
| | 5 = University degree, college degree or above | 3 | 1.2% |
| Career | 1 = Farmer | 120 | 48.8% |
| | 2 = Others | 126 | 51.2% |
| Understory economy production mode | 0 = Understory breeding | 81 | 32.9% |
| | 1 = Understory planting | 165 | 67.1% |

In China's rural households, the farming labor force accounts for a large proportion, and the farmer household livelihood sources are relatively single. The main economic source of most surveyed farmer households was understory economic industry. Only 9% of the rural households were village cadres with a higher education level. The distance between the home and the forest land was mostly around 1–3 km, and more than 30% of the forest land was transferred. The surveyed households had been engaged in the understory economy for an average of 5–7 years, and the total annual income of the understory economy accounted for more than 50% of the total annual income of the family. Hence, the economic benefits brought by the understory economy were relatively large (see Table 5 for details).

**Table 5.** Individual characteristics of farmer households and forest land management status.

| Variable | Average Value | Standard Deviation | Median | Minimum | Maximum |
|---|---|---|---|---|---|
| The proportion of farming labor to household labor | 64.48% | 21.44% | 60.00% | 20.00% | 100.00% |
| Village cadres at home | 0.09 | 0.28 | 0.00 | 0.00 | 1.00 |
| Whether forest land is transferred | 0.31 | 0.47 | 0.00 | 0.00 | 1.00 |
| The average distance of forest land from home | 1.76 | 0.65 | 2.00 | 1.00 | 3.00 |
| The working time in understory economy | 6.60 | 5.19 | 5.00 | 1.00 | 40.00 |
| Proportion of understory economic income | 56.09% | 35.66% | 66.67% | 0.00% | 100.00% |

- Environmental perception

Overall, the farmer households showed a good overall cognition of environmental protection, with average scores larger than 4. However, regarding the different perceptions, there were great differences in the cost perception and responsibility perception of farmers' pro-environmental behaviors, whereas the differences in other aspects of perception were small. This indicates that farmers generally agree with the concepts that the environment needs to be protected and that environmental protection is beneficial to sustainable human development. However, due to the influences of cost and other factors, they often try to circumvent their environmental protection responsibilities, showing a deviation of practice from knowledge (see Table 6 for details).

**Table 6.** Descriptive statistics of farmers' environmental perception.

| Variable | Average Value | Standard Deviation | Median | Minimum | Maximum |
|---|---|---|---|---|---|
| $P_1$ | 4.43 | 0.75 | 5.00 | 2.00 | 5.00 |
| $P_2$ | 4.36 | 0.79 | 5.00 | 1.00 | 5.00 |
| $P_3$ | 4.36 | 0.78 | 5.00 | 2.00 | 5.00 |
| $P_4$ | 4.34 | 0.97 | 5.00 | 1.00 | 5.00 |
| $P_5$ | 4.29 | 0.74 | 4.00 | 2.00 | 5.00 |
| $P_6$ | 4.19 | 1.07 | 5.00 | 1.00 | 5.00 |

- Social constraints

The standard deviation of the scores of the farmer households' reputation appeal was 0.83, which is relatively large. The pursuit of social prestige is more subjective due to the influences of personal age, experience, and notion. However, the rural society of China is still dominated by the differential pattern with genetic relationships as links, parent notions, and group pressure have strong constraints on farmer households, and the mean value was, therefore, larger than 4 (see Table 7 for details).

**Table 7.** Descriptive Statistics of Social Constraints.

| Variable | Average Value | Standard Deviation | Median | Minimum | Maximum |
|---|---|---|---|---|---|
| $S_1$ | 3.77 | 0.83 | 4.00 | 2.00 | 5.00 |
| $S_2$ | 4.21 | 0.72 | 4.00 | 2.00 | 5.00 |
| $S_3$ | 4.05 | 0.75 | 4.00 | 2.00 | 5.00 |
| $S_4$ | 3.78 | 0.75 | 4.00 | 2.00 | 5.00 |
| $S_5$ | 4.04 | 0.76 | 4.00 | 2.00 | 5.00 |
| $S_6$ | 4.14 | 0.76 | 4.00 | 1.00 | 5.00 |

- Government incentives

The influences of government incentives can be analyzed from the two aspects of ecological compensation and government guidance. The average value of ecological compensation was 0.52, indicating that the government of the surveyed region pays considerable attention to ecological protection behaviors in terms of the understory economy. However, regarding guidance, both the mean and median values from the perspective of professional technical training of understory economy were close to 3, indicating that the frequency of technical training was about "once half a year". Via the survey, we also found that a large number of farmers engaged in the understory activities mainly rely on their own to learn from the experience of other participants. The amount of technical training was insufficient. Although the government also organized professional training, the frequency was too low to meet the development needs of farmers (see Table 8 for details).

**Table 8.** Descriptive statistics of Government incentives.

| Variable | Average Value | Standard Deviation | Median | Minimum | Maximum |
|---|---|---|---|---|---|
| $G_1$ | 0.52 | 0.50 | 1 | 0 | 1 |
| $G_2$ | 2.89 | 1.31 | 3 | 1 | 5 |

## 5. Results and Discussion

### 5.1. Results

#### 5.1.1. Regression Result Analysis

Prior to data analysis, we first conducted a multicollinearity test on the variables. As the inflation variance factors of all variables were below 10, there was no serious multicollinearity among variables. Further, the factors influencing the pro-environmental behaviors of farmer households in the understory economy were explored by ordinal probit regression and ordinal logit regression models to synchronously verify the stability and reliability of the results. Table 9 shows the regression results.

**Table 9.** Regression results.

| Dependent Variable: Level of Pro-Environmental Behaviors | | Oprobit (1) | Ologit (2) |
|---|---|---|---|
| Environmental perception | $P_1$ | 3.494 *** (0.619) | 6.899 *** (1.359) |
| | $P_2$ | 1.202 *** (0.285) | 2.370 *** (0.573) |
| | $P_3$ | 1.781 *** (0.393) | 3.425 *** (0.788) |
| | $P_4$ | 0.930 *** (0.269) | 1.689 *** (0.520) |
| | $P_5$ | 1.601 *** (0.322) | 3.192 *** (0.694) |
| | $P_6$ | 1.993 *** (0.325) | 3.978 *** (0.736) |
| Social constraints | $S_1$ | 1.470 *** (0.338) | 2.865 *** (0.675) |
| | $S_2$ | 1.010 *** (0.323) | 1.813 *** (0.610) |
| | $S_3$ | 0.881 *** (0.293) | 1.888 *** (0.599) |
| | $S_4$ | 1.228 *** (0.289) | 2.605 *** (0.651) |
| | $S_5$ | 0.236 (0.286) | 0.475 (0.544) |
| | $S_6$ | 1.148 *** (0.337) | 2.275 *** (0.690) |
| Government incentives | $G_1$ | 1.468 *** (0.470) | 2.720 *** (0.899) |
| | $G_2$ | 0.100 (0.138) | 0.122 (0.267) |
| Individual characteristics | Age | 0.403 (0.265) | 0.722 (0.503) |
| | Gender | 1.214 * (0.640) | 2.320 * (1.195) |
| | Education | 0.957 *** (0.268) | 1.861 *** (0.547) |
| | Career | 0.904 *** (0.254) | 1.754 *** (0.496) |

**Table 9.** *Cont.*

| Dependent Variable: Level of Pro-Environmental Behaviors | | Oprobit (1) | Ologit (2) |
|---|---|---|---|
| | The proportion of farming labor to household labor | 1.936 ** (0.894) | 4.073 ** (1.795) |
| | Village cadres at home | −0.472 (0.662) | −1.022 (1.235) |
| Forest land management status | Whether forest land is transferred | 1.377 *** (0.446) | 2.821 *** (0.905) |
| | The average distance of forest land from home | 0.339 (0.249) | 0.593 (0.448) |
| | The working time in understory economy | 0.128 *** (0.050) | 0.245 *** (0.098) |
| | Proportion of understory economic income | 2.098 *** (0.541) | 4.211 *** (1.111) |

Standard error in brackets; *, **, and *** Significant at 10%, 5%, and 1%, respectively.

Overall, 19 variables had significant positive influences, among which 17 were significant at the 1% level. The results of ordinal probit regression and ordinal logit regression showed consistent significance, and the regression results were robust and reliable.

From the perspective of core variables, all variables used to measure environmental behavior had a positive significance. The stronger their environmental perception, the more likely farmers were to adopt pro-environmental behaviors. Among the variables used to measure social constraints, only the key people benefits failed the significance test. Most likely, environmental awareness has become a consensus to some extent from the perspective of the farmers themselves. This is consistent with the conclusion in the discussion of the core variables of environmental protection perception. Except for the variable of key people benefit, all other variables were significant at the 1% level. Regarding the variables used to measure government incentives, ecological compensation was significant at the 1% level; this factor has a strong guiding effect on farmers' behaviors. Driven by economic factors, farmers' behaviors can well reach an agreement with the policy objective of "ecological priority". However, the diversification of ecological compensation forms needs to be considered to achieve a good implementation effect [49].

From the perspective of control variables, the education level and the gender of farmers were significant at the levels of 1% and 10%, respectively. The higher the education level, the stronger the sense of sustainable development. Thus, limitations can be overcome better, and pro-environmental behaviors can be adopted. In terms of gender, due to the limited educational resources in rural areas and the restriction of traditional ideas, men are more likely to receive education than women and are, therefore, more inclined to consciously adopt pro-environmental behaviors in economic activities. From the perspective of forest land management, forest land transfer, the proportion of understory economy income, and the time of engaging in relative activities were significant at the 1% level. The longer the farmers engage in the understory economy and the higher the proportion of the income they can obtain, the higher their bond with the forest land and the deeper their affection, which makes them more likely to adopt pro-environmental behaviors. This phenomenon confirms the significance of occupation and concurrent occupation among the individual characteristics of farmer households. Regarding the forest land transfer factor, most of the small households in the surveyed area transferred their own forest land out to obtain rent and received labor remuneration from the large households or enterprises that acquired land for farming. By contrast, large households attached importance to the comprehensive benefits of large-scale forest land management and understory economy. Hence, the factor of forest land transfer was also significant at the 1% level. The more concentrated the forest land, the more likely the farmer households were to adopt pro-environmental behaviors.

### 5.1.2. Marginal Effect Analysis

Based on the analysis of the influencing factors, the influence degrees of these factors on pro-environmental behavior were further explored, and the marginal effects of the core variables were measured. As seen in Figure 3, the influence interval of the main factors influencing farmers' pro-environmental behaviors was from −0.2 to 0.15. The inflection point appeared between Behavioral Level 2 (relatively low pro-environmental behavior level) and Behavioral Level 3 (relatively high pro-environmental behavior level). The trends of the core influencing factors were consistent, and the behavior directivity was clear.

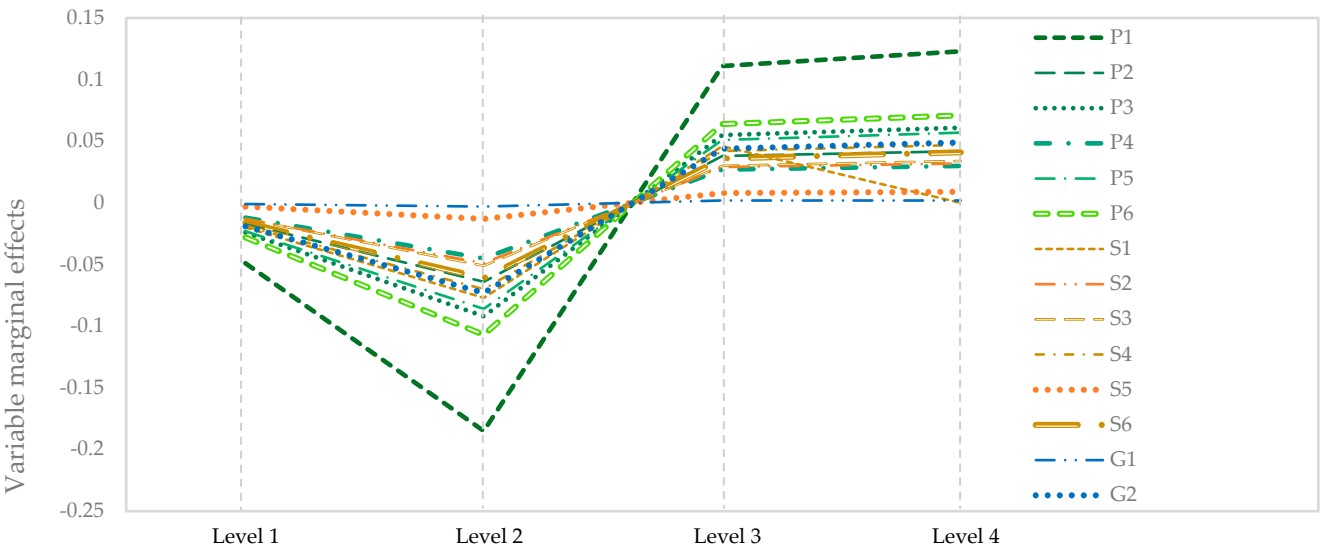

**Figure 3.** Relationship between marginal effect of variables and level of farmer households' pro-environmental behaviors.

Since the marginal values of each factor at Level 2 were negative and those at Level 3 were positive, this paper continues to discuss and analyze the marginal effect of each factor at these two levels (results shown in Table 10).

All indicators of environmental perception were significant at the 1% level. The top three factors were farmers' environmental awareness, perception bias of pro-environmental behaviors, and perceptions of benefits. With economic development and social progress, Chinese farmers have progressively improved their awareness of environmental protection. They gradually realize that environmental protection can bring long-term benefits. For example, environmental improvement is good for physical and mental health as well as economic development and can leave an "ecological heritage" for their children and grandchildren, thus producing greater comprehensive benefits. This is in line with the concept of "intergenerational equity" in the theory of sustainable development [50]. In this regard, if farmers' environmental awareness can be internalized into the self-drive force of pro-environmental behaviors, it can become a valuable spiritual motivation. However, if the cost of ecological protection is too high, it would be unfavorable for farmers to adopt pro-environmental behaviors [51].

Under social constraints, the top three factors that can improve the probability of pro-environmental behaviors are reputation appeal, group pressure, and social trust. The villages where farmer households live are generally "close-distance" societies. The behaviors of farmer households are often affected by the traditional concept of "individual prestige", and farmers are extremely concerned about the evaluation by others. They crave a high reputation in the group, expect to be trusted by their neighbors, and are more susceptible to group pressure. In contrast to urban residents, they rarely migrate and have a deep attachment to their hometown [52,53]. Moreover, they are easily influenced by their

parents' production and life management modes and generally transfer these modes to the next generation. "Pro-environmental behavior" is a positive behavior in public cognition. To "earn individual prestige", farmers will succumb to "group pressure" or subconsciously follow the good production and management advice of their parents, which makes them more likely to actively adopt pro-environmental behaviors [54].

**Table 10.** Variable marginal effects.

| Independent Variable | | Levels of Pro-Environmental Behaviors | | | |
|---|---|---|---|---|---|
| | | 1 | 2 | 3 | 4 |
| Environmental perception | $P_1$ | −0.049 *** (0.011) | −0.185 *** (0.037) | 0.111 *** (0.029) | 0.123 *** (0.017) |
| | $P_2$ | −0.017 *** (0.005) | −0.064 *** (0.013) | 0.038 *** (0.009) | 0.042 *** (0.009) |
| | $P_3$ | −0.024 *** (0.006) | −0.092 *** (0.021) | 0.055 *** (0.014) | 0.061 *** (0.012) |
| | $P_4$ | −0.012 *** (0.005) | −0.045 *** (0.011) | 0.027 *** (0.007) | 0.030 *** (0.009) |
| | $P_5$ | −0.023 *** (0.006) | −0.086 *** (0.018) | 0.051 *** (0.013) | 0.057 *** (0.010) |
| | $P_6$ | −0.028 *** (0.006) | −0.107 *** (0.020) | 0.064 *** (0.016) | 0.071 *** (0.010) |
| Social constraints | $S_1$ | −0.020 *** (0.005) | −0.077 *** (0.019) | 0.046 *** (0.014) | 0.051 *** (0.009) |
| | $S_2$ | −0.013 *** (0.005) | −0.049 *** (0.014) | 0.029 *** (0.008) | 0.032 *** (0.011) |
| | $S_3$ | −0.013 *** (0.005) | −0.051 *** (0.016) | 0.030 *** (0.011) | 0.034 *** (0.010) |
| | $S_4$ | −0.019 *** (0.005) | −0.070 *** (0.018) | 0.042 *** (0.013) | 0.047 *** (0.010) |
| | $S_5$ | −0.003 (0.004) | −0.013 (0.015) | 0.008 (0.009) | 0.009 (0.010) |
| | $S_6$ | −0.016 *** (0.006) | −0.061 *** (0.017) | 0.036 *** (0.011) | 0.041 *** (0.012) |
| Government incentives | $G_1$ | −0.001 (0.002) | −0.003 (0.007) | 0.002 (0.004) | 0.002 (0.005) |
| | $G_2$ | −0.019 *** (0.007) | −0.073 *** (0.024) | 0.044 *** (0.015) | 0.049 *** (0.016) |
| Individual characteristics | Education | −0.013 *** (0.004) | −0.050 *** (0.015) | 0.030 *** (0.010) | 0.033 *** (0.009) |
| | Career | −0.012 *** (0.004) | −0.047 *** (0.014) | 0.028 *** (0.009) | 0.031 *** (0.008) |
| | The proportion of farming labor to household labor | −0.029 ** (0.013) | −0.109 ** (0.049) | 0.065 ** (0.030) | 0.073 ** (0.032) |
| Forest land management status | Whether forest land is transferred | −0.020 *** (0.008) | −0.076 *** (0.023) | 0.045 *** (0.016) | 0.050 *** (0.015) |
| | The working time in understory economy | −0.002 ** (0.001) | −0.007 ** (0.003) | 0.004 ** (0.002) | 0.004 ** (0.002) |
| | Proportion of understory economic income | −0.030 *** (0.008) | −0.113 *** (0.031) | 0.068 *** (0.022) | 0.075 *** (0.016) |

Standard error in brackets; **, and *** Significant at 5%, and 1%, respectively.

Among government incentives, only the factor of government ecological compensation passed the significance test. With a per unit increase in government ecological compensation, the probability of farmers choosing the pro-environmental behaviors of Level 3 increased by 4.4%, whereas that of choosing the pro-environmental behaviors of Level 2 decreased by 7.3%. Government ecological compensation can directly increase farmer household income and has a direct influence on pro-environmental guidance. However,

government guidance did not pass the significance test. According to the field investigation, this is not only because the training frequency offered by the government is low but also because the training content is not in line with the actual needs of the farmers. Often, the training content is more theoretical than applicable, and some techniques often lag behind the actual development in the field. As a result, the training provided by the government often becomes a mere formality.

Regarding the personal characteristics of farmer households, if the education level of farmers increases by one unit, the probability of choosing the pro-environmental behaviors of Level 3 will increase by 3%, and that of choosing behaviors of Level 2 will decrease by 5%. This makes it necessary to pay attention to the specialized education oriented toward rural areas to effectively guide farmers that wish to adopt pro-environmental behaviors. In addition, the more professional the farmers are, the more likely they are to adopt such behaviors.

In terms of the management status of forest land, with a per-unit increase in the transfer of forest land, the probability of choosing the pro-environmental behaviors of Label 3 increased by 4.5%, and that of choosing behaviors of Level 2 decreased by 7.6%. Similarly, the probability of choosing the pro-environmental behaviors of Level 3 and Level 2 increased by 6.8% and decreased by 11.3%, respectively, when it came to the understory economy income. These results indicated the importance of an understory economy in farmer households and that the fragmentation of forest land affected the pro-environmental behaviors largely. Thus, long-term benefits and moderate scale of forest land management can effectively promote pro-environment behavior.

*5.2. Discussion*

There is no clear international concept of understory economy for the time being, and those with a more similar connotation to understory economy are social forestry, agroforestry, and non-wood forest products. Among them, agroforestry is the most similar to the concept of the understory economy in this study. At present, the comprehensive evaluation system of agroforestry is also maturing, and a large number of scholars have analyzed the efficiency of agroforestry [55,56]. However, in these assessments, economic and ecological benefits are given equal weight. China has a long history of using undergrowth space for planting and farming activities, but due to the public nature of land ownership, the development of the undergrowth economy model is permitted by state policy, and the responsible person for the forest land uses the undergrowth resources that can be developed to carry out related production and business activities. Particularly, in the development of the understory economy, unlike other models, ecological protection is given priority over economic functions. Currently, a large number of Chinese scholars have conducted qualitative studies and classifications on the development models of the understory economy [57,58]. These understory economy development models have been in the process of continuous refinement, presenting diverse and local characteristics. Depending on the climatic and geographic conditions, the understory economy has different development patterns in China. For example, in Yunnan province, people mainly grow mushrooms under forests; in Fujian province, people grow medicinal herbs under forests; in the northeast area, people grow ginseng under forests, etc.

Following the 20th National Congress of the Communist Party of China, Chinese people are striving together to build a modern socialist country and promoting the great rejuvenation of the Chinese nation through Chinese-style modernization. Such modernization is to obtain both common prosperity for all people and the harmonious coexistence between humans and the natural. It is obvious that the ecological environment must be considered an important factor in realizing sustainable and high-quality development. Based on the current development of China's understory economy, this study identifies the factors influencing farmers' pro-environmental behavior and explores their marginal contributions, providing useful references for governments that want to both improve farmers' income and espouse the concept of sustainability. Compared with previous research, the

innovation of this paper lies in the combination of theoretical analysis and field research to explore the factors influencing farmers' pro-environmental behaviors in understory activities. The results were highly targeted and effective in this field.

The direct motivation for the farmers' participation in the understory economy is to increase their income. According to the "limited rationality" of farmer households, the blind pursuit of economic benefits may cause ecological damage and is inconsistent with the concept of sustainable development. In contrast to previous studies, we highlight that under the current development level of China's understory economy, the environmental awareness of farmer households has a higher probability of influencing the level of pro-environmental behaviors. When the cost of pro-environmental behaviors is greater than the benefit, farmers tend to avoid pro-environmental behaviors [59]. Hence, some mandatory measures should be taken to promote the adoption of such behaviors. When the cost is lower than the benefit, perceptual factors and informal institutions are more effective in prompting farmers to adopt pro-environmental behaviors [60]. There is no doubt that economic incentives have a positive impact on farmer behavior, but they may also have a negative impact on the adoption of pro-environmental behaviors by farmers. This is particularly the case when economic incentives are interrupted, and farmers tend to wait for economic stimulus, which results in the phenomenon of "motivation crowding" [61]. In this regard, we should pay attention to the diverse forms of government ecological compensation to stimulate the internal motivation of farmers [62,63].

However, as the data were collected during the global COVID-19 pandemic, they were limited, and if we could conduct farm surveys on a larger scale, more sample data would make the results more convincing. In terms of the selection of research regions, not all major forestry provinces in China have been considered, such as the Guizhou Province in the southwest of China, and future studies should further conduct inter-regional comparisons and exploration. Moreover, based on the field investigation, we found that in recent years, moderate-scale production has become the trend of Chinese rural economic development, so the small farmer households mostly subcontracted their forest land to large households, which led to the relatively limited participation of individual farmers in the understory economy. In this regard, we should further study the pro-environmental behaviors of emerging new forestry business agents, such as agroforestry cooperatives and leading enterprises.

## 6. Conclusions and Suggestions

*6.1. Conclusions*

First, on the whole, environmental perception, social constraints, and government incentives all have significant influences on farmers' pro-environmental behaviors in understory economic activities. In addition, the heterogeneity of individual characteristics of farmers, including gender, education level, occupation, and farming labor force proportion, as well as the forest land management factors such as forest land transfer and the proportion of understory economy income, also influence the decision to adopt pro-environmental behaviors.

Second, from the perspective of the relationship between the marginal benefits of influencing factors and the behavioral levels, the probability contributions of the influencing factors to improving the level of farmers' pro-environmental behaviors are environmental perception, social constraints, and government incentives. Among the government incentive factors, few indicators passed the significance test, which may be due to the deviation in understanding and implementing related policies of local governments, leading to unsatisfactory effects. According to the results of field surveys and interviews, the fact that the government fails to consider the actual needs of farmers when providing guidance may be the main reason, especially in the aspects of policy publicity as well as green technique training and application.

Third, although the economic benefit is the main incentive for the decision making of farmer households, perception factors (environmental perception) and informal constraints

(social constraints) play a great role in the choice of famers to adopt pro-environmental behaviors, even though they are not mandatory.

In conclusion, different farmers make different behavioral choices. Some specific variables increase the probability that farmers choose a level of pro-environmental behavior. Non-economic factors, especially environmental perceptions and social constraints, should be taken into account when guiding farmers to adopt pro-environmental behaviors in the process of understory economic development.

*6.2. Suggestions*

First, strengthen publicity and education in terms of environmental responsibility awareness. In the vast rural areas, governments should focus on cultivating an awareness of responsibility for the environment. The government should accurately grasp the psychological anchor point of farmers, stimulate their emotional attachment to the place where they grew up, and awaken their awareness of environmental responsibility. In addition, farmers should be made aware of the long-term benefits brought by environmental protection and of the "intergenerational relations" to spread the concept of sustainable development. The reasonable use of social constraints is helpful to form a benign social norm and produce a positive pressure conducive to ecological, environmental protection, thus promoting farmers to adopt environmental protection behaviors.

Second, accelerate the establishment and improvement of ecological reward-and-punishment mechanisms. To promote the adoption of pro-environmental behaviors, it is necessary to make rational use of reward-and-punishment mechanisms through the combination of material and mental reward and punishment. Fine examples and models of pro-environmental behaviors should be set up and give full play to these positive models. Advanced individuals, squads, and groups should be rewarded hierarchically. On the other hand, negative models should also be set up for behaviors that destroy the ecological environment. Appropriate punitive measures, including but not limited to self-criticism, notified criticism, rectification, and fines, should be implemented. In addition, irregular inspections should be conducted by the local government, and mutual community-level supervision should be encouraged. The nomination of village cadres raised by villagers should be advocated.

Third, enhancing the training in pro-environmental production techniques. The frequency of such training should be increased, and fixed training points should be set up to implement regular training. The forms of training should be more varied, such as online and offline training, one-to-one, one-to-many, and many-to-many modes. The actual production needs of the farmers should be considered more, and complex techniques should be taught in a language that is easy to understand. Moreover, trainers and training institutions should pay greater attention to the feedback from farmers and reasonably evaluate the training effect. In this sense, trainers should promote the long-term participation of farmers in relevant courses and frequently update the applied techniques.

In the face of climate change, the development of the understory economy will lead to sustainable economic and ecological benefits, and greater global attention should be paid to reducing the negative environmental impacts of production and management practices. While the above recommendations are specific to the current state of development of the understory economy in China, they are also applicable to developing countries with large populations, such as China, in their pursuit of balancing economic development and environment protection to improve the rural communities, resilience, and sustainability under the context of clime change.

**Author Contributions:** Conceptualization, Y.C. and H.L.; methodology, X.H. and S.L.; software, X.H. and S.L.; validation, Y.C., X.H. and H.L.; formal analysis, Y.C. and X.H.; investigation, X.H., S.L. and B.S.; resources, H.L. and Y.C.; data curation, X.H., S.L. and X.Z.; writing—original draft preparation, Y.C., X.H., S.L. and B.S.; writing—review and editing, Y.C., X.H. and X.Z.; visualization, Y.C. and X.H.; supervision, Y.C., X.H. and H.L.; project administration, Y.C. and H.L.; funding acquisition, Y.C. All authors have read and agreed to the published version of the manuscript.

**Funding:** This research was supported by the National Forestry and Grassland Administration Project of "Inclusive finance to support rural forestry development: A case study of Fujian Province" (grant number 0012906) and the Beijing Forestry University Project of "Study on the behavior motivation and guiding mechanism of forest farmers under the goal of high quality development of understory economy" (grant number 2021SRY10).

**Institutional Review Board Statement:** Not applicable.

**Informed Consent Statement:** Informed consent was obtained from all subjects involved in the study.

**Data Availability Statement:** The data presented in this research are available on request from the corresponding author.

**Acknowledgments:** The authors are particularly grateful to all researchers and institutes for providing data for this study. The authors are also very grateful to the editors and reviewers for their comments and suggestions for improving this study.

**Conflicts of Interest:** The authors declare no conflict of interest.

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
