# Peer review of "The Influencing Factors of Pro-Environmental Behaviors of Farmer Households Participating in Understory Economy: Evidence from China"

_sustainability, doi:10.3390/su15010688_

Round 1

Reviewer 1 Report

Dear authors

I read the manuscript entitled “The Influencing Factors of Pro-environmental Behaviors of Farmer Households Participating in Understory Economy: Evidence from China”. This manuscript is of suitable topic and has focused on an important issue. It has great potential to be published in Sustainability and I strongly recommend it for publication. However, the authors should try to address the following minor comments before its consideration for publication.

General comments

1.       The share of results should be increased in the abstract section. In other words, please try to mention some more results in the abstract.

2.       Please also highlight one of the critical contributions of present study to the body of knowledge.

3.       In my opinion, should be contextualized in China in a more concrete manner. Please mention that why this study should be done in China? This should be clearly highlighted with reference to some reliable sources. This can improve the problem statement within the manuscript.

4.       The main objective of the study should be divided into some operational sub-objectives.

5.       The main question of the study should be added to the manuscript.

6.       The discussions and recommendations should be put in an international scope. In other words, please try to go beyond the boundaries of China in representation and comparison of the results and discussions.

7.       Please try to focus on the broad impacts of the study in the discussion section.

8.       Some of the most important limitations of the study should be highlighted in the end of conclusion section.

9.       The main take-home message of the study should be presented in the form of one single or two sentences in the end of conclusion section.

Specific comments

1.       In page 1 (Lines 39-42), the authors have mentioned that “Nevertheless, in contrast to agroforestry, understory economy is not an utterly industrial operation mode but emphasizes the principle of "ecological priority", which is a unique idealized development concept proposed by the Chinese government”. This argument should be supported with reliable sources.

2.       In page 2 (Lines 50-55), the respected authors have mentioned that “The purposes of developing understory economy are to promote economic growth, improve the livelihood of forest farmers, consolidate the achievements of poverty alleviation, make up for the inherent shortcomings of forestry, such as long operation cycles, and obtain economic and social benefits on the premise of ensuring the ecological benefits of forests without destroying their ecological resources.”. These claims should be references.

3.       In page 2 (Lines 84-88), the authors have mentioned that “On the one hand, environmentally friendly behaviors can reduce the cost of environmental governance, and on the other hand, they can facilitate a healthy development of the understory economy industry. Therefore, it is of great importance to guide farmer households to take environmental protection behaviors when participating in understory economy.” Again, this argument should be supported with some sources.

4.       In page 2 (Lines 89-91), the respected authors have mentioned that “However, the specific factors that affect farmer households' pro-environmental behaviors when participating in understory economy and how these factors affect the pro-environmental behaviors of farmer households are still unclear.” However, I believe that such a claim should be made with caution. Because I have recently read some great articles from different researchers in the world who have worked on the factors affecting the environmental behavior of farmers. I believe that these articles should be reviewed, summarized, and cited in this section and the differences between your research and these researches should be stated. This can make your research problem and the contribution your work has made to the body of knowledge more concrete.

-       Chao, S. H., Jiang, J. Z., Wei, K. C., Ng, E., Hsu, C. H., Chiang, Y. T., & Fang, W. T. (2021). Understanding pro-environmental behavior of citizen science: An exploratory study of the bird survey in Taoyuan’s farm ponds project. Sustainability13(9), 5126.

-       Burke, J., & Running, K. (2019). Role identities and pro-environmental behavior among farmers. Human écology review25(1), 3-22.

-       Xie, H., & Huang, Y. (2021). Influencing factors of farmers' adoption of pro-environmental agricultural technologies in China: Meta-analysis. Land use policy109, 105622.

-       Valizadeh, N., Esfandiyari Bayat, S., Bijani, M., Hayati, D., Viira, A. H., Tanaskovik, V., ... & Azadi, H. (2021). Understanding Farmers’ Intention towards the Management and Conservation of Wetlands. Land10(8), 860.

5.       In page 3 (Section 2.1 and Lines 105-150), the respected authors have explained the research framework. I believe that this section should not be in methodology section. Therefore, it should be moved before the methodology section (AFTER INTRODUCTION). The mentioned sources in the comment 4 can be used to enrich the research framework as well.

6.       Methodology section has been written and elaborated very well.

7.       Result section is also presented and articulated very well.

8.       The discussion should be put in an international scope. This can improve the applicability and attractiveness of the manuscript for the international readers.

9.       In page 18, I believe that the limitations of the study should be highlighted.

In general, I believe that this manuscript can be accepted for publication in Sustainability after Minor revisions.

Author Response

Dear Professor,

  Best regards.

Reviewer 2 Report

Dear authors,

I have found your paper very interesting. The paper of understory economy is crucial in the current global context of climate-change. It will increase the resilience of both rural and urban municipalities, as well as it will promote local economies. The methodology and structure of your paper is proper. The only thing I think it is essential to include in the discussion section is a link of the understory economy with the concept of green infrastructures, as well as a paragraph in the conclusion section regarding the limitations of your research.

Author Response

Dear Professor,

  Best regards.
